# PLoRA: Efficient LoRA Hyperparameter Tuning for Large Language Models

## Abstract

Low-rank Adaptation (LoRA) has gained popularity as a fine-tuning approach for Large Language Models (LLMs) due to its low resource requirements and good performance. While a plethora of work has investigated improving LoRA serving efficiency by serving multiple LoRAs concurrently, existing methods assume that a wide range of LoRA adapters are available for serving. In our work, we conduct extensive empirical studies to identify that current LoRA training paradigms do not utilize hardware resources efficiently and require high overhead to obtain a performant LoRA adapter. Leveraging these insights, we propose PLoRA, which automatically orchestrates concurrent LoRA fine-tuning jobs under given hardware and model constraints and develops performant kernels to improve training efficiency. Our experimental studies show that PLoRA reduces the makespan of LoRA fine-tuning over a given hyperparameter search space by up to $7.52\times$ and improves training throughput by up to $12.8\times$ across a range of state-of-the-art LLMs.

## 1 Introduction

Large Language Models (LLMs) have become the backbone of numerous modern AI applications, spanning natural language understanding, code generation, multimodal reasoning, and specialized domains such as healthcare and finance Guo et al. (2025); Grattafiori et al. (2024); Yang et al. (2024). The paradigm of pre-training followed by fine-tuning has enabled models to achieve state-of-the-art performance when adapted to specific tasks Ouyang et al. (2022).

However, fine-tuning large models for multiple tasks or user-specific applications poses a significant challenge due to the high computational cost of training and serving numerous fine-tuned variants. To address this, parameter-efficient fine-tuning (PEFT) techniques such as Low-Rank Adaptation (LoRA) Hu et al. (2021a) have emerged as scalable alternatives to full fine-tuning. LoRA significantly reduces the number of trainable parameters by introducing low-rank decomposition matrices into Transformer layers, allowing for specialization while keeping the pre-trained model weights frozen. This versatile deployment approach has become popular, and several systems have been recently developed to serve multiple LoRA adapters concurrently Sheng et al. (2023); Chen et al. (2024).

Existing LoRA-related inference systems, such as vLLM Kwon et al. (2023), SLoRA Sheng et al. (2023), and LoRAX Zhao et al. (2024), operate under the assumption that LoRA adapters are already well-trained and a LoRA checkpoint with decent model quality is available for a given downstream task. In this paper, we focus on the question of *how to train such LoRA adapters efficiently*.

Similar to other deep learning methods, we find that the effectiveness of LoRA fine-tuning hinges on selecting appropriate hyperparameters (§2.2). In the context of LoRA, hyperparameter tuning extends beyond standard parameters, such as learning rate and batch size. LoRA-specific parameters that need to be tuned include: 1. LoRA rank, which controls the dimensionality of the adapter matrices. A higher rank increases the expressive power but comes at a higher memory and computation cost Hu et al. (2021a). 2. A scaling factor $\alpha$, which determines the impact of the LoRA adapters on the pre-trained weights. We conduct a large-scale empirical study and demonstrate that there is no single rule of thumb for tuning LoRA hyperparameters. Through more than 1,000 experiments, we demonstrate that different tasks (e.g., mrpc Wang et al. (2018) or gsm8k Cobbe et al. (2021)) require different hyperparameter configurations to achieve optimal performance on various base models, highlighting the need for LoRA hyperparameter tuning.

However, traditional hyperparameter tuning techniques Kandasamy et al. (2020); Bergstra and Bengio (2012); Bergstra et al. (2011) only focus on reducing the number of tuning runs and do not account for the unique characteristics of LoRA adapters, leading to significant inefficiencies. Unlike conventional fine-tuning, where each configuration is trained in isolation, we aim to enable *multiple LoRA adapters to be trained concurrently within the same run*. We refer to this setting as **intra-run** training. Since LoRA adapters are **heterogeneous** and have **varying resource requirements** (§2.3), intra-run settings open up new optimization opportunities. For example, we find that many LoRA configurations evaluated during hyperparameter tuning have small batch sizes and underutilize GPU hardware resources, with SM occupancy around $16.7\%$ and memory utilization less than $55\%$. Based on these observations, we propose *packing multiple LoRA configurations* during hyperparameter tuning, thereby sharing hardware resources across configurations and improving utilization.

We develop PLoRA, an automated concurrent LoRA training system. Given a base model and a predefined hyperparameter search space, PLoRA orchestrates efficient LoRA fine-tuning jobs and executes them with packed LoRA adapters. PLoRA operates in two stages: an offline planning stage followed by an online execution stage. We first design an offline packing planner that analyzes the hyperparameter search space and creates jobs with packed LoRA configurations that maximize throughput. The planner also determines the appropriate degree of parallelism for each job. We formulate the planner as an optimization problem and design an efficient approximate algorithm with provable performance bounds (§5).

In the second stage, the jobs created by the planner are deployed by an online LoRA Execution Engine. The execution engine monitors available hardware resources and launches multiple tuning jobs concurrently if sufficient resources are available. We also design new GPU kernels for packed LoRA adapters that are used once a job is launched, and show that our kernels can achieve near-linear speedups for up to 32 adapters across various base models. In summary, we make the following contributions:

- We conduct a large-scale empirical study with more than 1,000 experiments to demonstrate the necessity of optimizing LoRA hyperparameters.
- We develop an algorithm to automatically allocate multiple LoRA fine-tuning tasks to jobs while accounting for hardware, model, and LoRA configuration constraints.
- We demonstrate that PLoRA reduces the total hyperparameter tuning makespan by up to $7.52\times$ when tuning more than 100 configurations.

## 2 HYPERPARAMETER TUNING IN LoRA

### 2.1 LOW-RANK ADAPTATION (LoRA)

Low-Rank Adaptation (LoRA) Hu et al. (2021a) is a widely adopted efficient fine-tuning technique for Large Language Models (LLMs). Formally, for a weight matrix $W \in \mathbb{R}^{d \times k}$, LoRA adds the weight updates $\Delta W$ as two small matrices $A \in \mathbb{R}^{d \times r}$ and $B \in \mathbb{R}^{r \times k}$, where $r$ is the LoRA rank and much smaller than $d$ and $k$. The additional FLOPs incurred by LoRA is linear to its rank. LoRA only updates $A$ and $B$, thus significantly reducing computation and storage costs for fine-tuning.

During inference, LoRA merges the multiplied matrix $\Delta W = B \times A$ into the original weight matrix $W$ with a scaling factor LoRA alpha $\alpha$ and the weight matrix becomes $W = W + \alpha \Delta W$, as shown in Figure 1.

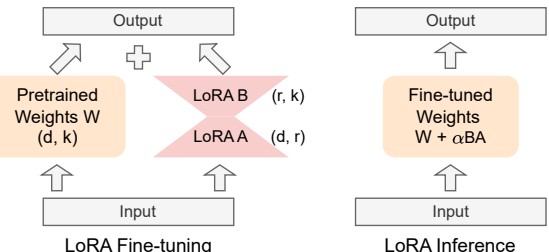

Figure 1: This figure demonstrates how LoRA is applied to weight matrices in fine-tuning and inference.

### 2.2 HYPERPARAMETER SPACE

Hyperparameter tuning is a fundamental process in developing deep learning models, involving selecting optimal values for parameters not learned during training Bergstra et al. (2011). These hyperparameters include, but are not limited to, the *learning rate*, which determines the step size for updating model weights, and the *batch size*, which defines the number of samples processed before a weight update. LoRA fine-tuning also requires hyperparameter tuning and introduces additional hyperparameters, such

Table 1: Hyperparameters for LoRA fine-tuning. LoRA alpha is a scaling factor on LoRA updates.

| Hyperparameters | Search range |
|---|---|
| Learning rate (LR) | $2\text{e-}5 \sim 4\text{e-}4$ |
| Batch size (BS) | $1 \sim 32$ |
| LoRA rank (r) | $8 \sim 128$ |
| LoRA alpha ($\alpha$) | $r/4 \sim 4r$ |

Table 2: This table shows the optimal hyperparameter configuration we found during the hyperparameter sweep.

| | 3B | | | | 7B | | | |
|---|---|---|---|---|---|---|---|---|
| Task | Rank | LR | BS | $\alpha$ | Rank | LR | BS | $\alpha$ |
| mrpc | 16 | 4e-5 | 1 | 1 | 32 | 6e-5 | 1 | 1 |
| cola | 64 | 4e-4 | 1 | 0.25 | 32 | 8e-5 | 1 | 0.5 |
| wnli | 32 | 2e-4 | 2 | 1 | 32 | 2e-4 | 4 | 0.5 |
| gsm8k | 32 | 1e-4 | 2 | 1 | 16 | 3e-4 | 1 | 1 |

Table 3: This table analyzes LoRA hyperparameter sensitivity. The base model is QWen-2.5-7B. We only tune one hyperparameter and keep the others fixed. The results are the maximum accuracy differences by tuning the chosen hyperparameter.

| Task | LR | BS | Rank | $\alpha$ |
|---|---|---|---|---|
| mrpc | 8.5% | 10.0% | 6.4% | 4.9% |
| cola | 14.2% | 8.5% | 13.1% | 5.9% |
| wnli | 6.8% | 11.3% | 5.4% | 5.5% |
| gsm8k | 5.0% | 3.2% | 4.5% | 2.5% |

Table 4: Model quality of QWen-2.5-7B with the base model only, the worst, and the best LoRA hyperparameter configuration across various tasks. $\Delta$ represents the accuracy improvements between the best configuration and the base model.

| | Base | Worst | Best | $\Delta$ |
|---|---|---|---|---|
| mrpc | 64.1% | 57.1% | 70.0% | +5.9% |
| cola | 62.7% | 61.5% | 80.2% | +18.5% |
| wnli | 78.8% | 74.7% | 84.5% | +5.7% |
| gsm8k | 70.8% | 71.2% | 79.8% | +9.0% |

as *LoRA rank* and *LoRA alpha*, which are specifically related to LoRA adapters. The set of hyperparameters studied in this paper for LoRA fine-tuning is listed in Table 1. Introducing these LoRA-specific hyperparameters expands the search space of hyperparameter tuning, making the process more complex and challenging.

## 2.3 STUDY OF LoRA HYPERPARAMETER TUNING

We perform an extensive empirical study of the impact of hyperparameters on model quality with LoRA fine-tuning. The detailed experimental setup is described in §6. We use QWen-2.5 Yang et al. (2024) as the base model and report the zero-shot accuracy on the following benchmarks in our experiments: **GSM8K** Cobbe et al. (2021) for mathematical reasoning; **mrpc** Wang et al. (2018) for language understanding; **cola** Wang et al. (2018) for commonsense reasoning; and **wnli** Wang et al. (2018) for logic reasoning.

**Observation #1: Hyperparameters strongly influence LoRA model quality.** We investigate both individual and collective impacts of hyperparameters on model quality. First, by varying only one hyperparameter at a time (Table 3) while fixing others to the optimal configuration in Table 2, we find accuracy differences of up to $14.2\%$ (learning rate), $11.3\%$ (batch size), $13.1\%$ (LoRA rank), and $5.9\%$ (LoRA $\alpha$). Second, by evaluating 120 LoRA configurations (Table 4) built from the search space in Table 1, we observe that hyperparameters collectively have a substantial effect: while some configurations degrade accuracy below that of the pre-trained base model (e.g., wnli drops from $78.8\%$ to $74.7\%$), careful tuning can yield significant improvements (e.g., cola $+18.5\%$, wnli $+5.7\%$, gsm8k $+9.0\%$).

We also study how the best LoRA configurations vary across different LoRA fine-tuning workloads, which are evaluated using both QWen-2.5-3B and QWen-2.5-7B as base models. The best LoRA configurations for different workloads are listed in Table 2.

**Observation #2: Optimal LoRA configurations vary across tasks and base models.** Table 2 shows that the best hyperparameter settings for LoRA fine-tuning depend on both the downstream task and the base model. For instance, with QWen-2.5-3B, the best configuration for mrpc is [16, 4e-5, 1, 1], while gsm8k requires [32, 1e-4, 2, 1]; applying the mrpc configuration to gsm8k reduces accuracy by $7.4\%$. Similarly, transferring the best configuration for cola on QWen-2.5-7B to QWen-2.5-3B decreases accuracy by $3.6\%$. These results highlight that effective LoRA fine-tuning requires workload- and model-specific configurations.

**Observation #3: LoRA fine-tuning benefits from small batch sizes.** As shown in Table 2, LoRA consistently achieves higher accuracy with smaller batch sizes ($\leq 4$), a trend also reported in prior work Zhao et al. (2024); Hu et al. (2021a); Fomenko et al. (2024). Smaller batches reduce gradient variance when only a fraction of parameters are updated, which improves convergence and generalization Hu et al. (2021a).

## 3 EFFICIENT LoRA HYPERPARAMETER TUNING

Next, we investigate the system efficiency of LoRA fine-tuning. We first discuss systems challenges in searching for the best LoRA configuration in a large search space. We then propose a new LoRA fine-tuning paradigm that significantly optimizes throughput for LoRA hyperparameter tuning, and discuss some new systems challenges that arise.

### 3.1 HARDWARE UNDERUTILIZATION

**SM underutilization.** We profile the SM occupancy for single LoRA fine-tuning on an A100 GPU with QWen-2.5-7B on Unsloth Daniel Han and team (2023). While we vary batch size from 1 to 16 and rank from 8 to 128, the SM occupancy remains constant at $16.7\%$ for both base model and LoRA kernels. This constant low occupancy arises because LoRA's much smaller matrices lack the arithmetic intensity and shared memory data reuse needed to fully utilize the tensor cores, suggesting that most SM resources sit idle during adapter updates.

**Memory underutilization.** When a single LoRA configuration is fine-tuned for a given set of hardware resources, GPU memory is often also underutilized. This underutilization arises from two factors: 1) The base model remains frozen, so only the relatively small LoRA adapter is updated, and 2) LoRA fine-tuning typically uses small batch sizes, further reducing memory demand.

**Existing hyperparameter tuning approaches fall short.** Existing hyperparameter tuning techniques, such as grid search Bergstra et al. (2011), and Bayesian optimization Kandasamy et al. (2020), focus on reducing the number of tuning runs and are agnostic to the time taken or resources required for each configuration. Thus, when applying these approaches to LoRA fine-tuning, the hardware resource is still underutilized during each LoRA configuration run.

### 3.2 OUR PROPOSAL: LoRA FINE-TUNING WITH PACKED LoRA CONFIGURATIONS

To address the above challenges, we propose packing multiple LoRA configurations for simultaneous fine-tuning, thereby sharing the same hardware resources to improve hardware utilization.

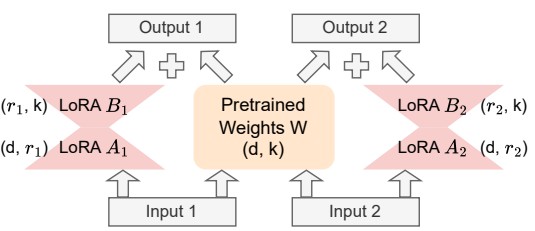

**Packing LoRA adapters is feasible.** Each LoRA configuration corresponds to a distinct LoRA adapter, while the base model remains identical across all configurations, as its weights are frozen during LoRA fine-tuning. This insight motivates the idea of packing multiple LoRA configurations into a single fine-tuning job, thereby avoiding out-of-memory (OOM) issues. When performing tensor parallel (or

Figure 2: This figure demonstrates how PLoRA works with two LoRA adapters packed in a fine-tuning job.

FSDP-based Zhao et al. (2023)) distributed fine-tuning, the combined memory capacity increases, allowing more LoRA adapters to be packed into a single fine-tuning job.

**Packed LoRA fine-tuning workflow.** In LoRA fine-tuning with a single LoRA adapter Hu et al. (2021a), a single input is passed to both the base model and the LoRA adapter, and their outputs are merged with a scaling factor $\alpha$ for the final output, as shown in Figure 1. In contrast, packed LoRA fine-tuning takes an array of multiple inputs with an input for each LoRA adapter, as shown in Figure 2. All inputs are passed to the base model and their corresponding LoRA adapters. The outputs from the base model and LoRA adapters are then merged. The computation of each adapter in packed LoRA fine-tuning is identical to that of LoRA fine-tuning with a single LoRA adapter. Meanwhile, the base model is shared among LoRA adapters, offering opportunities for higher hardware utilization.

### 3.3 CHALLENGES

Packing multiple LoRA adapters into a single fine-tuning job is analogous to increasing the batch size, improving hardware utilization and LoRA hyperparameter tuning efficiency, but introducing new challenges.

**Efficient computation of packed LoRA adapters on GPUs.** A packed LoRA fine-tuning job consists of a shared base model and multiple LoRA adapters. While the base model processes the combined input from all adapters, each adapter has distinct inputs and weights, preventing direct merging. Naively iterating over adapters, as in Figure 2, leads to low hardware utilization in both forward and backward passes due to small LoRA ranks and low arithmetic intensity.

**Resource-aware packed LoRA scheduling.** Even with optimized kernels, packing adapters efficiently is challenging because hardware cannot hold all configurations from a large hyperparameter search space. We must maximize utilization while avoiding OOM errors and allocate GPU resources across fine-tuning jobs. Maximizing throughput requires jointly optimizing both adapter packing and compute allocation; optimizing either in isolation is insufficient.

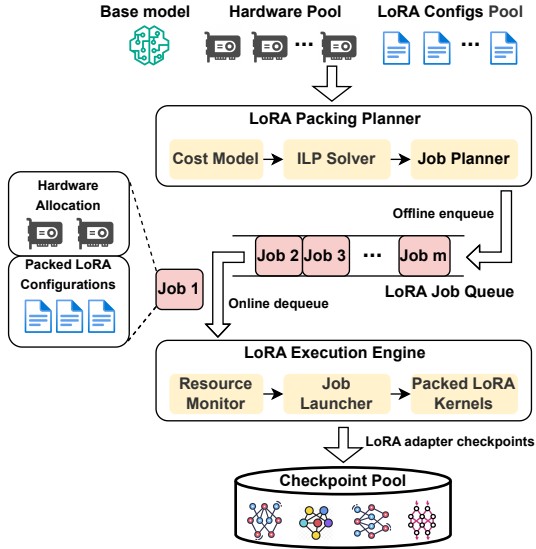

Figure 3: The system architecture of PLoRA.

## 4 SYSTEM OVERVIEW OF PLoRA

We propose PLoRA, a system for efficient LoRA hyperparameter tuning via packed fine-tuning. Given a base model, hardware pool, and configuration search space, PLoRA maximizes tuning throughput by packing LoRA configurations to fully utilize hardware resources. Figure 3 illustrates PLoRA, which has two main components: 1) a *LoRA Execution Engine* that launches packed fine-tuning jobs with optimized Packed LoRA Kernels (§C), and 2) a *LoRA Packing Planner* that schedules configurations by jointly optimizing packing and hardware allocation (§5).

PLoRA operates in two phases: *offline configuration planning* and *online execution*. A *fine-tuning job* is defined as fine-tuning multiple packed LoRA adapters on a shared base model. In the offline phase, the Packing Planner explores packed configurations using a cost model that estimates memory usage and throughput from the first few iterations (10 in our testbed). The Job Planner then determines packing strategies, allocates hardware resources, and enqueues planned jobs.

In the online phase, the Execution Engine dynamically dequeues jobs based on available hardware. Its Job Launcher sets up parallelism strategies and deploys packed jobs, while the Resource Monitor tracks resource availability. PLoRA can run multiple jobs concurrently as long as resources suffice. Customized Packed LoRA Kernels improve GPU utilization during both forward and backward propagation (Appendix C).

Upon job completion, each adapter is stored in the Checkpoint Pool, and the released hardware resources are returned to the pool. The Resource Monitor then triggers execution of the next queued jobs, ensuring continuous and efficient hardware use.

## 5 SCHEDULING OF PACKED LoRA FINE-TUNING

This section describes how to schedule packed LoRA configurations for LoRA hyperparameter tuning. We first formalize the optimization problem by jointly considering LoRA configuration packing and hardware allocation for fine-tuning jobs. Since the formulation is NP-complete, we develop an approximate algorithm for this problem and analyze its performance.

The optimization goal is to minimize the *makespan* of training time for all configurations in the given search space on the specified hardware. We observe that the completion time of a LoRA fine-tuning job is mainly affected by two factors: **1. The packed LoRA configurations**, which determine the set

of LoRA adapters fine-tuned in a job. **2. The degree of parallelism**, which determines the number of GPUs used for each fine-tuning job.

The optimization problem aims to minimize the makespan $t_{opt}$. The detailed formulation can be found in Appendix A. This optimization problem is NP-complete as it can be viewed as a variant of a 0-1 knapsack problem Karp (2009). Our goal is to solve the makespan problem (Eq. 8). We approximate this by maximizing instantaneous throughput (Eq. 1), which we solve with DTM (Alg. 1) and prove is near-optimal (Appendix D). In addition, our optimization formulation adds a new layer of complexity, as each job must first decide which LoRA configurations ($\mathcal{H}_{jk}$) to train and determine the associated degree of parallelism.

$$\max \sum_{j=1}^{m} \frac{\sum_{k=1}^{|K|} \mathcal{H}_{j,k} * r_k}{T(\mathcal{H}_{j,k}, d_j)}, \quad (1)$$

$$\text{s.t.} \quad M_{\text{base}} + \sum_{k=1}^{|K|} \mathcal{H}_{j,k} * M_{\text{lora},k} \leq C * M_{\text{gpu}} * d_j, \quad (2)$$

$$\forall 1 \leq j \leq m$$

$$\Sigma_j d_j \leq G, \quad 1 \leq j \leq m \quad (3)$$

$$1 \leq d_j \leq G, \quad d_j \in \{2^i \mid i \in \mathbb{N}\} \quad (4)$$

$$m \geq 1, \quad m \in \mathbb{Z} \quad (5)$$

Since the LoRA FLOP (floating-point operations of adapters, excluding the base model) is fixed given a hyperparameter search space, minimizing the makespan ($t_{opt}$) is equivalent to maximizing the average LoRA FLOP over this time. Thus, the optimization problem can be rewritten in throughput form as $\max \frac{\sum_{k=1}^{|K|} FLOP_k}{t_{opt}}$, where $FLOP_k$ denotes the FLOP of configuration $k$. Because solving this problem exactly is intractable, we next develop an approximation algorithm to address it.

Computing an optimal job schedule for average throughput is challenging, so we instead maximize *instantaneous throughput* at the job level. This leads to a scheduling algorithm with provable bounds relative to the optimal solution.

$$F(D, K) = \frac{\sum_{k=1}^{|K|} \mathcal{H}_k * r_k}{T(\mathcal{H}_k, D)}, \quad (6)$$

$$\text{s.t.} \quad M_{\text{base}} + \sum_{k=1}^{|K|} \mathcal{H}_k * M_{\text{lora},k} \leq C * M_{\text{gpu}} * D, \quad (7)$$

**Maximizing fine-tuning job throughput.** Given $G$ GPUs and a set of LoRA configurations $K$, we approximate minimizing makespan by maximizing throughput (Expression 1).

Note that we use LoRA rank in Eq (1) instead of LoRA FLOP by leveraging the linear scaling property of LoRA FLOP in rank (refer to §2.1). Here $r_k$ is the rank of configuration $k$, $m$ the number of concurrent jobs, $d_j$ the parallelism degree of job $j$, $M_{\text{base}}$ the base model memory, $M_{\text{lora},k}$ the LoRA memory, $M_{\text{gpu}}$ the GPU capacity, and $C \in (0, 1]$ a load factor. Constraints 2–5 enforce memory, GPU, and parameter ranges.

**ILP Solvable with determined parallelism degree.** We note that the problem is nonconvex because $T(\mathcal{H}_{j,k}, d_j)$ depends on $d_j$. However, since $d_j$ takes power-of-two values, we can enumerate the denominator. We thus solve a restricted ILP where parallelism is fixed at $D \leq G$. Here $\mathcal{H}_k$ indicates whether configuration $k$ is selected. We solve $F(D, K)$ recursively with a DTM algorithm (Alg. 1): for each $D$, the solver optimizes $F(D, K)$, then recursively applies ILP to the remaining subproblems. The recursion terminates when no GPUs remain or all configurations are scheduled. Among all candidate schedules $P$, the one with the minimum makespan is selected.

---

**Algorithm 1:** Decomposed Throughput Maximization (DTM)

**Input:** Number of GPUs $G$, LoRA configuration space $K$

**Output:** Scheduling policy, which is a set of packed LoRA configurations and their parallelism degrees.

---

```
1 def DTMHelper(g, P_tmp, K, P):
2   if g ≤ 0 or K = ∅ then
3     P ← P ∪ P_tmp ;
      return;
4   g' ← 2^⌊log₂ g⌋ ;        // Round down
    // d represents parallelism
       degree
5   foreach d ∈ {g', g'/2, ..., 1} do
      // Call Gurobi ILP solver
6     P_new, K_used ← F(d, K);
7     DTMHelper(g - d,
        P_tmp ∪ P_new, K - K_used);

8 def DTM(G, K):
9   P ← ∅ ;
10  DTMHelper(G, ∅, K, P) ;
11  return arg min{T(p)| p ∈ P} ;
```

**The job planner.** Algorithm 1 focuses on finding the best packing of LoRA configurations on the available GPUs to maximize concurrent throughput. It determines which configurations to train together and to what degree of parallelism. The job scheduling algorithm then takes these packed groups and decides the order in which they should run on the hardware, producing a complete job queue. Together, these two algorithms first determine the most efficient packing of configurations (Algorithm 1) and then schedule those packs over time (job scheduler) (Algorithm 2). This overall process approximately solves Problem (1), which is equivalent to minimizing the makespan, by ensuring that hardware is kept as fully utilized as possible throughout execution.

The core principle for the job planner is to schedule packed LoRA fine-tuning jobs with the maximum concurrent throughput whenever hardware resources are available. If there are available GPU resources for job scheduling (Line 4), it invokes DTM() introduced in Algorithm 1 to find the best set of packed LoRA fine-tuning jobs for these available resources (Line 5) and updates the remaining LoRA configurations (Line 7). The job planner also adds the set of jobs to the LoRA job queue (Line 8). It then predicts the next job completion event with the cost model and updates the number of available GPUs for the next round of job planning.

**Computation time of the job planner.** The running time of the job planner is negligible, especially considering this is for offline scheduling. Since solving each optimization instance takes less than a second and all recursive branches can be performed in parallel, we observe that the computation time of Algorithm 1 is within 10 minutes in our evaluation with 120 configurations on 8 GPUs(§6.2), less than 2.5% of the overall duration. In Appendix D, we prove the near-optimality of the scheduling algorithm.

---

**Algorithm 2:** The Job Planner

**Input:** Number of GPUs $G$, LoRA configuration space $K$

**Output:** LoRA job queue $Q$

---

1 $Q \leftarrow []$ ;
2 Initialize available GPUs $g_{avail} \leftarrow G$ ;
3 **while** $K \neq \emptyset$ **do**
4     **if** $g_{avail} > 0$ **then**
5        $P \leftarrow \mathrm{DTM}(g_{avail}, K)$;
6        **foreach** $p \in P$ **do**
7           $K \leftarrow K - p.\text{configs}$ ;   // Update configs
8        $Q.\text{append}(P)$;
9     Predict next job completion event;
10     Update $g_{avail}$;
11 **return** $Q$;

---

## 6 EVALUATION

In this section, we will demonstrate the effectiveness of PLoRA for LoRA hyperparameter tuning. Specifically, we will address the following questions: 1. Can PLoRA reduce the makespan of LoRA hyperparameter tuning? (§6.2) 2. Does PLoRA find better LoRA adapters that improve model quality? (§6.4) We also conduct detailed ablation studies to evaluate each component's performance.

### 6.1 EXPERIMENT SETUP

**Testbed.** We conduct experiments with a G5 and a P4d 24xlarge instance from Amazon EC2. The P4d instance has 8 A100 GPUs (40 GB) connected by NVLink for GPU-to-GPU communication. The G5 instance has 8 A10 GPUs (24 GB) connected by PCIe Gen4 for GPU-to-GPU communication.

**Models and tasks.** We conduct experiments on the Qwen 2.5 model family, one of the frontier open-weight model families that provides the most complete model size selections, including, but not limited to $3B$, $7B$, $14B$, and $32B$, and on LLAMA-3.2-3B and LLAMA-3.1-8B. We perform our evaluation in a zero-shot setting, following the prompting template in prior work Zhao et al. (2024). We use four downstream tasks, GSM8K Cobbe et al. (2021), mrpc Wang et al. (2018), cola Wang et al. (2018), and wnli Wang et al. (2018) and set sequence length to 1024.

**LoRA configuration selection.** In LoRA hyperparameter tuning, the search space is specified by users. The search space in our evaluations consists of four knobs: learning rate, batch size, LoRA rank, and LoRA alpha. Their ranges are listed in Table 1. We select a total of 120 LoRA configurations for the experiments.

**Baselines.** We compare PLoRA with sequential approaches for LoRA hyperparameter tuning, in which each LoRA fine-tuning job only evaluates one LoRA adapter. We consider two strategies for sequential approaches: *Min GPU*, which uses the minimum set of hardware that satisfies the memory constraints for each LoRA fine-tuning job and launches parallel jobs to fill all GPUs; and *Max GPU*, which uses the maximum number of devices within a GPU instance for each LoRA fine-tuning job, i.e., setting TP degree to 8 in our testbed. While we evaluate it with tensor parallelism, we believe the

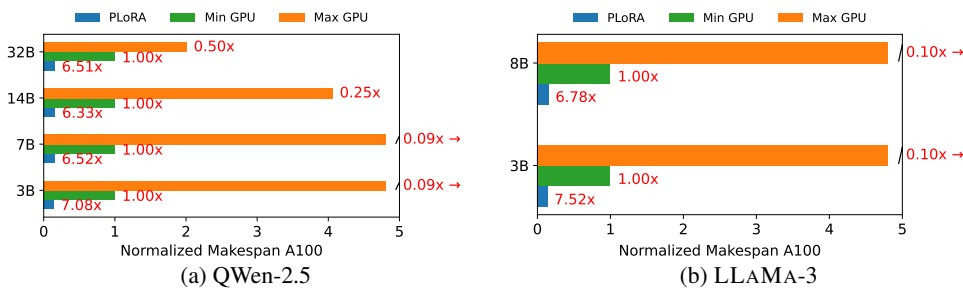

(a) QWen-2.5          (b) LLaMa-3

Figure 4: The makespan of LoRA hyperparameter tuning with different methods on A100 GPUs. The makespan is normalized to the performance of Min GPU.

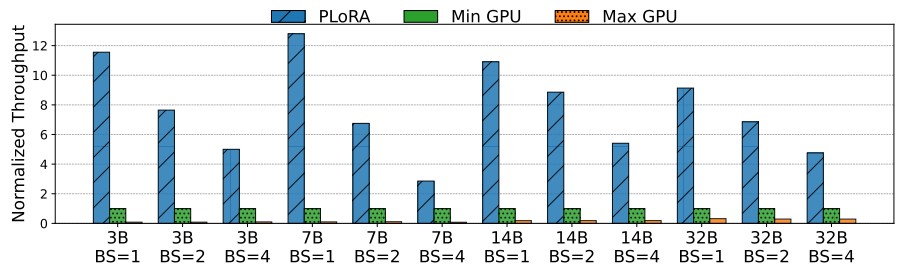

Figure 5: LoRA fine-tuning job throughput for various QWen-2.5 model sizes and batch sizes (BS) on A100 GPUs. The performance is normalized to Min GPU.

proposed design is also applicable to other parallelisms, such as pipeline parallelism Narayanan et al. (2019) and FSDP Zhao et al. (2023), which are part of our future work.

**Metrics.** We use the makespan to evaluate the end-to-end performance of LoRA hyperparameter tuning, which evaluates all LoRA configurations in the search space. We use throughput to evaluate the performance of LoRA fine-tuning jobs and packed LoRA kernels. We report the zero-shot accuracy on the downstream tasks.

**Implementation.** We implement a prototype of PLoRA atop torchtune torchtune (2024). Our implementation contains around 5000 lines of Python code and around 800 lines of code in a CUTLASS-based CUDA implementation for customized packed LoRA kernels. We use cvxpy Diamond and Boyd (2016) to implement our optimization module and built upon the PyTorch DTensor primitive to customize LoRA tensor parallel sharding strategies for efficient fine-tuning with tensor parallelism.

### 6.2 MAKESPAN EVALUATION.

We evaluate PLoRA's makespan improvement over two baselines using 120 LoRA configurations. The base models are from the QWen-2.5 family. A single GPU can fit the 3B and 7B models, while the 14B requires two GPUs and the 32B requires four. Thus, for Min GPU, we use this TP size and start concurrent jobs to fully occupy all available GPUs. For Max GPU, we use all eight GPUs per job, allowing only one job at a time.

Figure 4a shows the makespan normalized to Min GPU. The performance of Max GPU is much worse than that of Min GPU due to even lower hardware utilization. On the contrary, PLoRA reduces the makespan by $6.51\times$ and $6.33\times$ on 14B and 32B, respectively, thanks to packed LoRA fine-tuning. PLoRA also achieves $7.08\times$ and $6.52\times$ reduction in makespan on the 3B and 7B models.

We also evaluate LLaMa-3.2-3B and LLaMa-3.1-8B and observe similar improvements in makespan. Min GPU runs each LoRA fine-tuning job with one GPU, and eight concurrent jobs are launched. Max GPU still has the worst makespan, as shown in Figure 4b. PLoRA achieves $7.52\times$ speedups over Min GPU for LLaMa-3.2-3B and $6.78\times$ speedup for LLaMa-3.1-8B.

### 6.3 JOB-LEVEL THROUGHPUT EVALUATION.

We study the benefits of packing by measuring the throughput of packed LoRA fine-tuning jobs compared to our baselines. We run PLoRA with different base models and batch sizes on A100 GPUs. We fix LoRA rank to be 32 and other settings of PLoRA, Min GPU, and Max GPU are the

Table 5: Model quality comparison of different models with no LoRA, the default, and the best LoRA configuration. The numbers in each cell are the quality of the base model, the default configuration, the best configuration, and the quality improvement over the default configuration.

| | QWen-2.5-3B | QWen-2.5-7B | LLaMa-3.2-3B | LLaMa-3.1-8B |
|---|---|---|---|---|
| mrpc | 62.4 / 62.6 / 67.6 +5.0% | 64.1 / 64.7 / 70.0 +5.3% | 70.3 / 77.4 / 80.6 +3.2% | 71.3 / 80.3 / 84.5 +4.2% |
| cola | 48.8 / 53.8 / 77.2 +23.4% | 62.7 / 68.4 / 80.2 +11.8% | 69.9 / 71.8 / 77.3 +5.5% | 71.9 / 73.8 / 80.0 +6.2% |
| wnli | 53.5 / 66.2 / 73.4 +7.2% | 78.8 / 80.1 / 84.5 +4.4% | 46.4 / 61.9 / 64.8 +2.9% | 54.9 / 67.6 / 73.2 +5.6% |
| gsm8k | 61.2 / 64.8 / 74.6 +9.8% | 70.8 / 72.1 / 79.8 +7.7% | 60.4 / 63.3 / 71.3 +8.0% | 69.6 / 70.5 / 78.0 +7.5% |

same as those in §6.2. We show the job throughput on QWen-2.5 models in Figure 5. We observe similar trends in the LLaMa-3 models.

For a batch size of 1, PLoRA achieves up to $12.8\times$ speedup across the tested models. When we increase the batch size, the performance gain reduces since the Min GPU strategy can better utilize the hardware. However, the table shows that we still achieve a significant throughput improvement for a batch size of 4. Further increasing batch sizes harms model quality, as discussed in §3.

### 6.4 MODEL QUALITY WITH PLoRA

In this section, we evaluate the model quality of the best LoRA adapter found by PLoRA from the given search space with 120 LoRA configurations. Four base models, QWen-2.5-3B, QWen-2.5-7B, LLaMa-3.2-3B, and LLaMa-3.1-8B are fine-tuned with LoRA on four downstream tasks.

The model quality results are shown in Table 5. Each cell reports four numbers: (1) the base model without LoRA, (2) the LoRA adapter fine-tuned with the default hyperparameters from Unsloth Daniel Han and team (2023), a popular LoRA framework, (3) the best LoRA adapter found in our search space, and (4) the quality improvement (in red) of the best configuration over the default one. The results show that default LoRA hyperparameters already improve quality over the base model on downstream tasks. However, they do not fully exploit LoRA's potential. After searching 120 configurations with PLoRA, the best LoRA adapters outperform the default configuration by up to 23.4%, with consistent gains across different model families.

## 7 RELATED WORKS

**LoRA-related systems.** Efficient LoRA serving has been extensively studied. DLoRA Wu et al. (2024), Punica Chen et al. (2024), and SLoRA Sheng et al. (2023) develop scheduling algorithms, optimized GPU kernels, and memory management techniques for multi-LoRA serving. However, these systems focus on LoRA serving and assume that LoRA adapters have been well-trained. A recent work Ye et al. (2023) develops a pipeline parallel strategy for LoRA training. In contrast, PLoRA optimizes the system efficiency of LoRA hyperparameter tuning to find the best LoRA adapter from a search space.

**Hyperparameter tuning.** Existing hyperparameter tuning techniques, such as grid search Bergstra et al. (2011) and Bayesian optimization Kandasamy et al. (2020), are designed to reduce the search space, which is orthogonal to our study. PLoRA can work with different hyperparameter tuning algorithms based on the configuration space provided to the planner. Many hyperparameter tuning systems Dunlap et al. (2021); Mai et al. (2020); Akiba et al. (2019); Kotthoff et al. (2017); Li et al. (2020) have also been proposed for model pretraining, however, these techniques and systems cannot address low hardware utilization when fine-tuning a single LoRA configuration. Our work focuses on leveraging underutilized hardware resources to sweep the hyperparameter search space.

**Job scheduling.** Makespan minimization is an extensively studied topic in generalized cluster job scheduling Narayanan et al. (2020); Xiao et al. (2018); Hu et al. (2021b). These prior works on generalized cluster scheduling assume jobs are predefined. However, in LoRA hyperparameter tuning, PLoRA's optimization module also determines how LoRA configurations are packed into each job and their degree of parallelism. We leverage additional information about LoRA configurations and hardware resources to optimize job scheduling and joint allocation of GPU resources.

## 8 CONCLUSION

This paper presents PLoRA, a system for efficiently tuning LoRA hyperparameters. We conduct an extensive empirical study to demonstrate the need for LoRA hyperparameter tuning and identify inefficiencies in current tuning pipelines. We leverage these insights to design a LoRA packing

planner and an execution engine and build a parallel LoRA fine-tuning framework. PLoRA improves the fine-tuning throughput by up to $12.8\times$ over traditional approaches by packing multiple LoRA adapters in a fine-tuning job and reduces makespan by up to $7.52\times$ across tested models.

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

# A    OPTIMIZATION FORMULATION

The optimization problem can be formulated as follows:

$$\min \quad t_{opt} \tag{8}$$

$$\text{s.t.} \quad t_{opt} \geq s_j + T(\mathcal{H}_{j,k}, d_j), \quad \forall j \in J \tag{9}$$

$$\sum_{j \in J} \mathcal{H}_{jk} = 1, \quad \forall k \in K \tag{10}$$

$$s_{j'} \geq s_j + T(\mathcal{H}_{j,k}, d_j) - M(1 - \mathcal{W}_{jj'}) + \mathcal{Z}_{jj'}M, \tag{11}$$
$$\forall j \neq j', \; j, j' \in J$$

$$s_j \geq s_{j'} + T(\mathcal{H}_{j',k}, d_{j'}) - M(1 - \mathcal{W}_{jj'}) + \mathcal{Z}_{jj'}M, \tag{12}$$
$$\forall j \neq j', \; j, j' \in J$$

$$\mathcal{Z}_{jj'} + \mathcal{Z}_{j'j} = \mathcal{W}_{jj'}, \quad \forall j \neq j', \; j, j' \in J \tag{13}$$

$$\mathcal{W}_{jj'} \leq \mathcal{X}_{ij}, \; \mathcal{W}_{jj'} \leq \mathcal{X}_{ij'}, \forall j \neq j', \forall i \tag{14}$$

$$\mathcal{W}_{jj'} \geq \mathcal{X}_{ij} + \mathcal{X}_{ij'} - 1, \forall j \neq j', \forall i \tag{15}$$

$$\mathcal{X}_{ij}, \mathcal{H}_{jk} \in \{0,1\}, \quad \forall i, \; \forall j \in J, \forall k \in K \tag{16}$$

$$\mathcal{Z}_{jj'}, \mathcal{W}_{jj'} \in \{0,1\}, \quad \forall j \neq j', \; j, j' \in J \tag{17}$$

$$s_j \geq 0, \forall j \in J; \quad 1 \leq i \leq G \tag{18}$$

In this formulation, we input the number of hardware devices $G$ and LoRA configurations $K$. The rest are variables that the optimization instances solve. $J$ represents the set of jobs, $\mathcal{X}_{ij}$ is a binary variable that equals 1 if job $j$ is assigned to device $i$, $\mathcal{H}_{jk}$ is a binary parameter that indicates whether LoRA configuration $k$ belongs to job $j$. $s_j$ is the start time of job $j$, $\mathcal{Z}_{jj'}$ is a binary variable that ensures job ordering where $\mathcal{Z}_{jj'} = 1$ if job $j$ precedes job $j'$ and does not overlap in time, $\mathcal{W}_{jj'}$ is a binary variable indicating that whether job $j$ and $j'$ share at least one device. We employ $M$ as an auxiliary large constant in the ordering constraints Ghirardi and Potts (2005). During optimization for minimal makespan, the optimization instance would output how LoRA configurations are assigned to jobs ($\mathcal{H}$), how jobs are assigned to devices ($\mathcal{X}$), and when jobs are scheduled ($\mathcal{Z}$). $T()$ is the cost model used to estimate the training time of LoRA fine-tuning jobs; it is not a variable, but a function of the packed LoRA configurations $\mathcal{H}_{j,k}$ and the parallelism degree $d_j$, where $j$ is a job. The notations used in this section are listed in Table 6.

In our optimization setup, Equation equation 10 ensures that each LoRA configuration belongs to exactly one fine-tuning job; Inequalities equation 11, equation 12, and Equation equation 13 ensure that jobs sharing any devices do not overlap in time Pinedo (2008); Inequalities equation 14 and equation 15 help enforce the prior constraint by setting $\mathcal{W}_{jj'}$ to 1 when two jobs share at least one device Pinedo (2008). The makespan is represented as the latest job completion timestamp in Inequality equation 9. Equations equation 16, equation 17, and equation 18 define the scope of the variables. This optimization problem also has constraints on GPU memory usage of each LoRA fine-tuning job and the number of GPUs that can be allocated to all concurrent jobs.

This optimization problem is NP-complete as it can be viewed as a variant of a 0-1 knapsack problem Karp (2009). In addition, our optimization formulation adds a new layer of complexity, as each job must first decide which LoRA configurations ($\mathcal{H}_{jk}$) to train and determine the associated degree of parallelism.

## A.1    NOTATION TABLE

Below is the notation table used in formulating the optimization problem:

# B    LoRA MEMORY CONSTRAINTS

## B.1    LoRA MEMORY CONSTRAINTS

To ensure that the training job does not exhaust the available GPU memory, we impose a constraint on the total memory used by all LoRA configurations in a job, including the base model weights. The

Table 6: Notation used in cost model formulation

| Symbol | Description |
| --- | --- |
| **Model-related parameters** | |
| $C$ | Memory load factor |
| $M_{\text{base}}$ | Memory required for the base model |
| $M_{\text{lora},k}$ | Memory required for LoRA configuration $k$ |
| $M_{\text{gpu}}$ | Total memory capacity of a GPU |
| $G$ | Number of available GPU devices |
| $K$ | Set of LoRA configurations |
| $J$ | Set of training jobs |
| $M$ | A large constant for scheduling ordering |
| $T()$ | Duration of job, estimated with cost model |
| **Optimization variables** | |
| $\mathcal{X}_{ij}$ | Binary variable: 1 if job $j$ runs on device $i$ |
| $\mathcal{H}_{jk}$ | Binary variable for LoRA assignment |
| $s_j$ | Start time of job $j$ |
| $r_k$ | LoRA rank of LoRA configuration $k$ |
| $\mathcal{Z}_{jj'}$ | Binary variable for scheduling order |
| $\mathcal{W}_{jj'}$ | Binary variable for device sharing |
| $d_j$ | Number of GPUs used by job $j$ |

total memory usage must not exceed the GPU memory:

$$M_{\text{base}} + \sum_{k=1}^{K} M_{\text{lora},k} \leq c_{load} \times M_{\text{gpu}}$$

$M_{\text{base}}$ represents the memory cost of the base model, while $M_{\text{lora},k}$ represents the memory cost of a LoRA adapter on a device. Similar to other work on managing GPU memory Kwon et al. (2023), the user can set a load factor $c_{load}$ to account for internal GPU fragmentation and adjust GPU memory usage.

For each LoRA configuration $k$, the memory usage, $M_{\text{lora},k}$, is represented by an indicator variable to determine if the user applies LoRA to those matrices. In each attention block, the user can apply LoRA to $Q$, $K$, $V$, and the output projection matrix; In each MLP block, the user can apply LoRA to the up, down, and gate projection matrix. We, therefore, write the LoRA memory usage as a sum of these 7 components:

$$M_{\text{lora},k} = \sum_{i=1}^{7} \mathbb{I}_i M_{\text{lora},i} \tag{19}$$

Each index $i$ corresponds to one of the seven components the user can apply LoRA to. For each component, the LoRA memory usage includes the memory required to store LoRA parameters, gradients, and activations:

$$M_{\text{lora},k} = M_{\text{lora\_param},k} + M_{\text{lora\_grad},k} + M_{\text{lora\_act},k} \tag{20}$$

The memory for LoRA parameters, $M_{\text{lora\_param},k}$, is given by:

$$M_{\text{lora\_param},k} = n_{\text{layers}}(h_{\text{in}} \times r_{\text{lora},k} \\ + h_{\text{out}} \times r_{\text{lora},k}) \times c_{\text{prec}}$$

In practice, this may change depending on memory-saving strategies such as activation checkpointing. Here, $n_{\text{layers}}$ is the number of layers, $h_{in}$ and $h_{out}$ represent the input and output dimensions of the projection matrix, which can take different values based on the model architecture and vary between attention and MLP blocks. $c_{\text{prec}}$ is the training precision, representing bytes per parameter.

The memory required for the gradients, $M_{\text{lora\_grad},k}$, is calculated as:

$$M_{\text{lora\_grad},k} = c_{grad} \times M_{\text{lora\_param},k} \times c_{\text{prec}}$$

$c_{grad}$ represents the scaling factor for storing gradient-related parameters. For example, this factor is three in the popular AdamW optimizer, representing momentum, velocity, and primary gradients.

Finally, the LoRA activation memory for each block, $M_{\text{lora\_act},i}$, is given by:

$$M_{\text{lora\_act},k} = b \times s \times r_{\text{lora},k} \times c_{\text{prec}}$$

Here $b$ is the batch size, and $s$ is the sequence length. This term represents the memory used to store intermediate activations during training. In LLM fine-tuning, the sequence length varies based on the workload. The standard practice is to set a maximum training length and split the training document if some data samples are too long to ensure no memory overflow. When computing memory consumption, we take the same approach and set the sequence length to the maximum length of the training samples.

Similarly, the activation memory for the base model can be computed by summing the activation of the embedding layer, the attention operator, and the feed-forward network in each layer. Depending on the implementation, other activations, such as those produced by layer norm may also be computed and stored, our model can be adapted to those implementations with ease. More details on computing the memory consumption for each of these four modules can be found in the Appendix:

$$M_{\text{base\_act}} = M_{\text{base\_emb}} + M_{\text{base\_attn}} + M_{\text{base\_mlp}}$$

The total memory for the base model is then calculated as:

$$M_{\text{base}} = M_{\text{base\_weights}} + M_{\text{base\_act}}$$

The computation in this section assumes a single-device setting. The section discusses how the parallelization strategy affects our memory constraints.

### B.1.1 Parallelization strategy and GPU constraints:

The prior section of constraints assumes that a full copy of the model is stored on each device. In practice, tensor parallelism, pipeline parallelism, and fully-sharded data parallelism (FSDP) are popular strategies to parallelize LLM training and are necessary for modern LLMs. The following section explains how we incorporate different parallelization strategies into our cost model. To accommodate tensor parallelism and pipeline parallelism, we can rewrite the memory cost associated with the LoRA adapters $i$ as follows:

$$M_{\text{lora\_param},k} = \frac{M_{\text{lora\_param},k}}{d_{\text{tp}} * d_{\text{pp}}}$$

This is similar to base model parameters and intermediate outputs. For FSDP, the model computes the following for different levels of ZeRO optimizers. For ZeRO-1, the LoRA memory includes both the unsharded gradient and parameter memory and the sharded optimizer state:

$$M_{\text{lora},k}^{(1)} = M_{\text{lora\_grad},k}^{(1)} + M_{\text{lora\_param},k} + \frac{M_{\text{opt},k}}{d_{\text{fsdp}}}$$

For ZeRO-2, the gradient memory also includes the gradient term:

$$M_{\text{lora, k}^{(2)}} = M_{\text{lora\_param},k} + \frac{M_{\text{lora\_grad},k} + M_{\text{opt},k}}{d_{\text{fsdp}}}$$

For ZeRO-3, the memory includes all fully sharded components:

$$M_{\text{lora},k}^{(3)} = \frac{M_{\text{lora\_param},k} + M_{\text{lora\_grad},k} + M_{\text{opt},k}}{d_{\text{fsdp}}}$$

Then, the model will solve for concurrent training jobs to launch and determine each job's parallelization strategy, ensuring that the total GPU usage does not exceed the GPU constraints.

We apply this formulation to every memory cost computation and add a total GPU constraint where the model will solve for the number of jobs to launch concurrently, as well as the tensor parallel degrees and LoRA adapter configurations to be packed on each job:

$$\Sigma_j d_j \leq G$$

## C  OPTIMIZING PACKED LoRA COMPUTATION

### C.1  INEFFICIENT COMPUTATION IN EXISTING FRAMEWORKS

LLM pre-training frameworks, such as Megatron-LM Shoeybi et al. (2019) and PyTorch Zhao et al. (2023), and LoRA fine-tuning frameworks, such as PEFT Mangrulkar et al. (2022) and Unsloth Daniel Han and team (2023), only support fine-tuning one configuration on a set of hardware at a time. Since LoRA adapters, when packed, share the same base model weights but have different adapter weights and inputs[1], a naive approach to support fine-tuning with packed LoRA adapters is to batch the computation of the base model and sequentially compute each LoRA adapter (Figure 2). However, this approach results in poor fine-tuning throughput due to low hardware utilization when computing each LoRA adapter.

We profiled the fine-tuning performance using the naive approach as described above. We use Qwen-2.5-7B as the base model and apply a single LoRA adapter with batch size 1 on an A100 GPU as the baseline. The iteration time increases by $10\%$ when the batch size is increased from 1 to 8. However, when we pack eight adapters into a fine-tuning job and each adapter has a batch size of 1, the naive approach worsens the iteration time by $3.6\times$ compared to single LoRA tuning due to low hardware utilization in LoRA adapter computations. We also observe similar performance when fine-tuning Qwen-2.5-14B with two A100 GPUs and Qwen-2.5-32B with four A100 GPUs using TP. This confirms our hypothesis that the performance bottleneck is the sequential computation of LoRA adapters rather than the batched computation in the base model.

### C.2  PACKED LoRA KERNELS

We devise custom CUDA kernels for PLoRA to efficiently batch the computation of LoRA adapters in both forward and backward propagations. We carefully tile the LoRA matrices and group gradient computations across multiple adapters to improve hardware utilization and handle load balancing for heterogeneous LoRA adapters.

Given a set of LoRA adapters, we concatenate the LoRA adapters into a tensor and design kernels which can compute forward and backward passes for all LoRA adapters. Our key insight in designing performant kernels is to tile the concatenated tensor along the sequence or hidden dimensions if possible. The sequence dimension consists of the input token sequence multiplied by the batch size. We avoid tiling along the LoRA rank dimension because the rank can be as small as 8, and sharding on the smaller dimension prevents GPUs from fully utilizing their compute resources.

If we denote the dimension of LoRA A by $(d, r)$ and the dimension of LoRA B by $(r, k)$ (Figure 1), we typically have $d >> r$ and $r << k$. Previous work on serving multiple LoRA adapters Chen et al. (2024) introduces a kernel that handles these two cases separately, by splitting the input dimension $d$ for LoRA A and the output dimension $k$ for LoRA B. While this is possible for the forward pass, backward propagation cannot simply reuse this strategy. When computing the activation gradients with respect to LoRA inputs, avoiding tiling over the rank dimension would require splitting the inner dimension for tiling. This strategy would require extra overhead in creating a scratch buffer for each tile, additional indices for bookkeeping, and extra synchronization and reduction steps for accumulating intermediate results, which would undermine the benefits of tiling over large dimensions.

In our work, we tackle the challenge of efficient backward propagation implementation and use the following strategy to obtain performant kernels.

**CUDA kernel design.** We built upon CUTLASS for our packed LoRA kernels. Four cases should be considered separately for the upstream weights and input gradients of LoRA A and LoRA B. Below, we outline our partitioning strategy for backpropagation in detail.

• Case 1, in which we compute the gradient for the weight of the LoRA B projection. We partition along the output dimension (k) for the LoRA B projection matrix to ensure that the gradient computation correctly associates the $i$th LoRA's slice with the corresponding input and output matrix slices. Tiling is done along the output dimension, and LoRA ranks $r_1$ and $r_2$ remain in each tile.

---

[1]We replicate the input tokens for each LoRA adapter at the beginning of each fine-tuning iteration.

- Case 2, in which we compute the gradient for the input of the LoRA B projection. We tile over the sequence dimension and the LoRA rank dimension of the upstream gradient, and reduce over the input hidden dimension.

- Case 3, in which we compute the gradient for the weight of the LoRA A projection. We tile over the sequence dimension of input activations and the output dimension of the upstream gradients, using LoRA rank as the reduction axis.

- Case 4, in which we compute the gradient for the input of the LoRA A projection. We tile along the upstream gradients' sequence and LoRA rank dimensions and use the concatenated LoRA rank dimension for reduction.

**Kernel performance tuning.** To achieve high kernel performance across different hardware setups, we tune the ThreadblockShape, WarpShape, and InstructionShape parameters in CUTLASS Thakkar et al. (2023) to optimize performance. While the optimal settings vary depending on both the underlying hardware architecture and the GEMM problem dimensions, we simulate workloads using model dimensions from widely used 3B and 7B models, as well as sequence lengths ranging from 512 to 2048. We set the InstructionShape to $(16, 8, 16)$ to match the tensor core instruction shape on Ampere GPUs. For WarpShape, we empirically found that $(64, 64, 32)$ yields the best throughput on the A100, while $(16, 64, 32)$ performs best on the A10 without triggering memory errors. Based on these warp shapes, we configure ThreadblockShape as $(128, 128, 32)$ on the A100 and $(64, 64, 32)$ on A10 to ensure compatibility with WarpShape.

## D   PROOF OF GREEDY SCHEDULING TAIL EFFECT

**Algorithm analysis.** Algorithm 2 performs optimally in a streaming setting with an unlimited number of jobs, as it consistently selects the job with the highest concurrent LoRA fine-tuning throughput. However, in our setting with a finite number of jobs, this approach can lead to a tail effect: the final jobs may not fully utilize all available hardware resources, resulting in suboptimal overall throughput compared to the optimal solution. We now bound this tail effect.

**Theorem D.1** (Bounded Tail Effect with Algo 2). *Let $J$ be a set of jobs scheduled on $G$ GPUs. Let $j \in J$ be the last job that uses $D$ GPUs. Let $T_{last}$ be the fine-tuning time of the last job and $F$ be the makespan based on the job planner's schedule. Then, the approximation ratio $(AR)$ of the job planner for the makespan optimization problem is upper bounded by: $AR \leq \frac{F}{F - T_{last} \cdot \frac{G-D}{G}}$.*

See Appendix D for the detailed proof. In practice, using experiment settings from §6, we find that PLoRA produces schedules with AR between 1.05 and 1.14.

*Proof.* We start by noting that before starting the last job, all $G$ GPUs are fully utilized by the definition of our greedy scheduling algorithm. Moreover, our algorithm offers a monotonicity condition: if a job using $x$ GPUs is scheduled, the next job in the optimal ordering requires no more than $x$ GPUs. This condition guarantees no bubbles between jobs in the fully loaded batches; the only underutilization occurs in the final batch.

Define the total GPU work as $W = \sum_{j \in J} x_j t_j$. Recall that $t_{\text{last}}$ denotes the processing time of the last job and the cumulative time of the fully utilized jobs before starting the last job be $F_{prev}$. Let $F = F_{prev} + t_{\text{last}}$ be the makespan of the job planner's schedule, and OPT be the makespan of an optimal schedule with full GPU utilization throughout. Then we can write:

$$W = F_{prev} \cdot G + t_{\text{last}}(G - D),$$

and the total makespan of the greedy schedule is

$$F = F_{prev} + t_{\text{last}}.$$

An optimal schedule (with full GPU utilization in every batch) must satisfy

$$\text{OPT} \geq \frac{W}{G} = F_{prev} + t_{\text{last}} \frac{G - D}{G}.$$

Table 7: The normalized throughput improvement of packed LoRA kernels over sequential LoRA computations. The first number in each cell represents the throughput speedup in the forward pass, while the second number represents that in the backward pass.

| Num. LoRA | 3B Attention $d = 2048$ | 3B MLP 11008 | 7B Attention 3584 | 7B MLP 18944 |
|---|---|---|---|---|
| 2 | 2.00x / 2.01x | 1.98x / 1.98x | 1.90x / 1.92x | 1.99x / 1.99x |
| 8 | 7.98x / 7.96x | 7.60x / 7.67x | 7.51x / 7.92x | 7.77x / 7.80x |
| 32 | 29.0x / 30.0x | 26.5x / 26.9x | 26.7x / 31.2x | 28.4x / 28.7x |

Thus, we can bound the extra time incurred due to the bubble in the last batch by

$$F - \text{OPT} \le [F_{prev} + t_{\text{last}}] - \left[ F_{prev} + t_{\text{last}} \frac{G - D}{G} \right] \tag{21}$$

$$= t_{\text{last}} \left( 1 - \frac{G - D}{G} \right) \tag{22}$$

$$= t_{\text{last}} \frac{D}{G} \tag{23}$$

This result quantifies the tail effect under asynchronous scheduling: the extra time is proportional to the fraction of idle GPUs for the last job. And we can obtain our final bound:

$$\frac{F}{\text{OPT}} \le 1 + \frac{t_{\text{last}} \cdot \frac{D}{G}}{F_{prev} + t_{\text{last}} \cdot \frac{G - D}{G}} \tag{24}$$

which can be simplified to

$$\frac{F}{\text{OPT}} \le \frac{F}{F - T_{\text{last}} \cdot \frac{G - D}{G}}.$$

$\square$

# E    ADDITIONAL EXPERIMENT RESULTS

## E.1    MICROBENCHMARKS

### E.1.1    PACKED LoRA KERNEL PERFORMANCE.

We examine the performance of our customized LoRA kernels in various workloads on A100 GPUs. Consider a LoRA tensor with a shape $[r, d]$, where $r$ is the LoRA rank and $d$ is the hidden dimension in the base model. We vary both $r$ and $d$ to evaluate the computational efficiency of the packed LoRA kernel. We first fix $r = 64$, set $d$ to different values based on the hidden dimensions in the Attention and MLP layers of QWen-2.5-3B and QWen-2.5-7B, and set the batch size to 1. We pack different numbers of LoRA computations into a kernel (ranging from 2 to 32) and compare the forward and backward computation performance with a sequential baseline.

Table 7 reports the throughput improvement normalized to the performance of the baseline. As we increase the number of packed LoRA computations from 2 to 32, our packed LoRA kernels exhibit close to linear speedups over the baseline in both forward and backward propagation. This trend holds for a wide range of hidden dimensions, from 2048 to 18944, as well as LoRA ranks, from 8 to 128.

In Table 8, we present kernel speed-up as we scale up LoRA forward and backward kernels on A10 GPUs.

### E.1.2    SPEEDUP BREAKDOWN.

PLoRA's performance gains mainly come from two components: optimized GPU kernels for efficient packed LoRA computations and near-optimal scheduling for packing LoRA configurations. This

Table 8: In this table, we show the throughput improvement of attention and MLP forward and backward LoRA kernels using sequence length 1024 on A10 GPUs as we scale up the number of concurrent LoRA adapters. The first number in each cell represents the throughput improvement (FLOP/s) of the forward pass, while the second number represents the backward pass.

| # LoRA Dim | 3B Attention 2048 | 3B MLP 11008 | 7B Attention 3584 | 7B MLP 18944 |
|---|---|---|---|---|
| 2 | 1.98x/1.98x | 1.9x/1.86x | 1.94x/1.97x | 1.98x/1.90x |
| 8 | 7.65x/7.55x | 7.52x/7.42x | 7.48x/7.4x | 7.44x/7.5x |
| 32 | 25.95x/26.09x | 25.87x/26.14x | 27.24x/26.45x | 26.78x/26.97x |

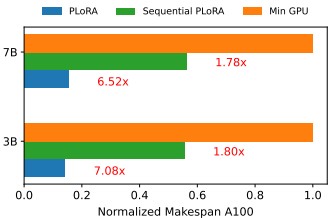

Figure 6: This figure shows the breakdown of PLoRA's speedup on A100 GPUs. Sequential PLoRA represents the speedup obtained via leveraging PLoRA's Packing Planner, but performs vanilla sequential LoRA training without PLoRA's Execution Engine.

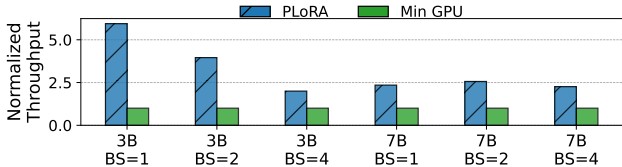

Figure 7: This figure shows the LoRA fine-tuning throughput for various models and batch sizes on A10 GPUs normalized to the Min GPU baseline.

ablation study breaks down how each component contributes to the overall reduction in makespan. We compare the makespan of LoRA hyperparameter tuning with Min GPU, using PLoRA for job planning but executing LoRA sequentially (Sequential PLoRA), and PLoRA. The results normalized to Min GPU are shown in Figure 6. We use QWen-2.5-3B and QWen-2.5-7B as base models and use a search space consisting of 120 LoRA configurations. Sequential PLoRA reduces the makespan by around $1.8\times$ for both models via amortizing the base model computation. The optimized GPU kernels further reduce the makespan by up to $3.93\times$, demonstrating that both components contribute significantly to PLoRA's performance.

### E.2 Fine-tuning Throughput on A10 GPUs

We evaluate PLoRA on A10 GPUs using the QWen-2.5-3B and QWen-2.5-7B models, with a LoRA rank of 32. The throughput of LoRA fine-tuning jobs is shown in Figure 7, and the performance is normalized to the Min GPU baseline. PLoRA achieves $5.94\times$ speedup for 3B and $2.56\times$ speedup for 7B. The throughput improvement is lower than that on A100 GPUs, which is expected because A10 GPUs have less GPU memory capacity than A100 GPUs and, therefore, can pack fewer LoRA adapters in LoRA fine-tuning jobs.

We also evaluate PLoRA on base models with QLoRA Dettmers et al. (2023), which quantizes the weights of the base model to 4 bits. QLoRA reduces the GPU memory usage of the base model, leaving more memory for LoRA adapters. We enable QLoRA in PLoRA and evaluate the performance with QWen-2.5-7B. We use LoRA with a rank of 32 and a batch size of 1 in all LoRA configurations. PLoRA achieves $4.72\times$ speedup compared to standard QLoRA fine-tuning with a single LoRA. This experiment shows that quantization, an orthogonal approach to boost LoRA fine-tuning efficiency,

can work with PLoRA to further improve fine-tuning throughput by packing more LoRA adapters in LoRA fine-tuning jobs.

## F  LLM USAGE

We used an LLM to polish the writing by correcting grammar in our completed draft. The LLM was not used to survey related work or to propose research ideas.

