# OpenReview forum: "PLoRA: Efficient LoRA Hyperparameter Tuning for Large Language Models"
_ICLR.cc/2026/Conference — Submitted to ICLR 2026_

### Official Review · Reviewer_wd2r · 2025-10-15

**Soundness:** 3
**Presentation:** 2
**Contribution:** 3
**Rating:** 6
**Confidence:** 3

**Summary:**

The paper presents PLoRA, a system for efficient hyperparameter tuning of Low-Rank Adaptation (LoRA) for large language models (LLMs). Unlike prior work focusing on multi-LoRA inference serving, PLORA tackles inefficiencies in LoRA training, especially during hyperparameter sweeps. The key idea is to pack multiple LoRA adapters with distinct hyperparameter settings into a single fine-tuning job, leveraging shared frozen base models to improve GPU utilization.

**Strengths:**

1. Novel problem framing.
The work identifies an under-explored inefficiency: hyperparameter tuning for LoRA adapters, which has received little systems-level attention compared to LoRA inference. The idea of intra-run concurrent training for multiple LoRAs is well-motivated.

2. Solid systems contribution.
PLORA provides a principled optimization formulation (NP-complete but approximated with an ILP-based algorithm) and a modular architecture with a packing planner, execution engine, and GPU kernels.

3. Substantial empirical gains.
The results are clear and significant: up to 7× shorter tuning time and 12× throughput improvement without quality loss across diverse models and tasks (MRPC, CoLA, GSM8K, WNLI). The experiments are thorough, covering hardware setups, baselines (Min/Max GPU), and sensitivity analyses.

4. Insightful empirical study.
Before introducing PLORA, the authors systematically show that LoRA hyperparameters (rank, α, batch size, LR) have strong and task-dependent effects, justifying the need for efficient tuning

5. Clarity and completeness.
The system design is clearly illustrated with well-labeled figures and pseudocode. The appendices further include a detailed cost model and memory constraints.

**Weaknesses:**

1. Limited comparison to existing hyperparameter tuning frameworks.
Although the authors position PLORA as orthogonal to Bayesian optimization and other search strategies, some empirical comparison could be helpful.

2. Fairness of baselines.
The “Min GPU” and “Max GPU” baselines are simple but may not represent the state of the art in distributed hyperparameter tuning. Showing how PLORA interacts with these would strengthen the systems claim.

3. Evaluation scope.
The experiments, though extensive, focus only on medium-scale models (up to 32B). It remains unclear how PLORA scales to >70B or multi-node settings, especially given communication overheads in packed LoRA kernels.

4. Theoretical clarity.
While the optimization formulation is detailed, the paper would benefit from a concise intuitive explanation of the trade-offs (e.g., how packing degree affects convergence noise or training interference). Currently, it’s heavy on equations but light on intuition.

**Questions:**

See weaknesses.

---

> ### Author Response · Authors · 2025-11-21
>
> We thank the reviewer for the review and recognize that our problem framing is novel, our systems contribution is solid, our empirical gain is significant and insightful, and that our work is clear and complete. We address your concerns below and hope this can earn your full support for the paper.
>
> **(W1) Comparison to HPO frameworks**
>
> As we stated in Related Works, PLoRA is orthogonal to, and not a replacement for, HPO search strategies like Bayesian Optimization (BO) and Hyperband [1]. PLoRA can serve as an accelerator for HPO strategies that solves the problem: "Given a set of N configurations, how do we execute them in the minimum amount of time?" Before our work, the answer was "run them one by one," which (as we show in Section 3) is extremely inefficient. PLoRA provides the first high-performance engine for this task.
>
> For instance, BO selects configs to try. PLoRA executes them. Before PLoRA, a parallel BO algorithm suggesting N configs would require N separate jobs. With PLoRA, the BO algorithm can submit all N configs to our planner, which can pack them into job(s), obtain the results, and feed them back to the BO much faster. PLoRA enables users to run much larger search spaces or for BO to explore more broadly, leading to better models.
>
> [1]. Hyperband: A Novel Bandit-Based Approach to Hyperparameter Optimization. Li et. al. https://arxiv.org/pdf/1603.06560
>
> **(W2) Fairness of baselines**
>
> MinGPU is exactly how a practitioner would approach this problem today. Users start a LoRA fine-tuning procedure using the minimum number of GPUs that does not overflow the memory.
>
> **(W3) Evaluation scope (up to 32B) and scalability to >70B / multi-node**
>
> Supporting LoRA across multiple nodes is an infrastructure challenge in itself. Current frameworks do not support multi-node training out of the box. For 70B models, the inefficiencies we identified in 3B to 32B models still hold. Memory is underutilized, where not all GPU memory is used. Similarly, hardware inefficiencies due to low arithmetic intensity in LoRA updates also persist. Therefore, our techniques would address these same issues at >70B scales.
>
> **(W4) Theoretical clarity (lacks intuition on trade-offs).**
>
> We thank the reviewer for the suggestion. The key trade-offs are:
>
> 1. Packing Degree vs. Throughput: Adding LoRA configurations to the same job adds minimal memory (just the adapter) but doubles the LoRA-related compute (FLOPs). This compute gain continues until the job becomes compute-bound or memory-bound (from all the packed adapters). Our ILP finds this optimal packing degree.
>
> 2. Parallelism (more GPUs) vs. Packing (more LoRAs): For a fixed pack of LoRAs, adding more GPUs decreases its runtime but reduces the number of other jobs we can run concurrently. Our DTM algorithm (Alg 1) explicitly searches this trade-off by iterating over possible parallelism degrees and finding the best combination of packing and parallelism that maximizes overall throughput.

---

> > ### Comment · Reviewer_wd2r · 2025-11-23
> >
> > Thank the authors for their response. I will keep my score.

---

### Official Review · Reviewer_nryQ · 2025-10-25

**Soundness:** 3
**Presentation:** 2
**Contribution:** 3
**Rating:** 6
**Confidence:** 2

**Summary:**

The paper presents PLORA, a system designed to drastically reduce the time and computational cost of hyperparameter tuning for RA fine-tuning. Its core innovation is the "packing" of multiple, heterogeneous LoRA configurations into a single training job, thereby improving hardware utilization. The work also demonstrates the advantage of using small batch sizes in LoRA fine-tuning, which makes the motivation more plausible.

**Strengths:**

1. The approach that packs LoRA configurations to improve hardware utility in hyperparameter tuning makes sense to me.
2. The empirical speedup in training time is impressive.

**Weaknesses:**

1. Although the authors have claimed through several observations that LoRA fine-tuning sometimes benefits from small batch sizes or configurations, I believe the most straightforward way to prove PLoRA's efficiency is to compare with baselines such as using larger ranks or batch sizes to improve hardware utilization. A lack of direct comparison with baselines like this somehow makes me unsure about whether the proposed techniques are essential.
2. I am not clear about the total efficiency gain of PLoRA. It seems that hyperparameter tuning usually costs a small fraction of time for the total training procedure. Taking the complete training cost after hyperparameter selection into consideration, I'm afraid PLoRA's efficiency gain in hyperparameter selection phase can be ignored. Maybe I'm wrong, but I expect to see clearer explanations on this issue.

**Questions:**

1. Can the authors give more explanations on the overall gain of PLoRA when considering the total training procedure, including both hyperparameter tuning and normal training?
2. There has been a lot of LoRA variants in the literature that can have better fine-tuning performance. I wonder if the core mechanism of PLoRA can also be extended to some successful LoRA variants?

---

> ### Author Response · Authors · 2025-11-21
>
> We thank the reviewer for the review and recognize that the empirical results are impressive. We address your concerns below and hope this can solidify your confidence in the paper and earn your full support.
>
> **(W1) Compare with baselines, such as larger ranks or batch sizes.**
>
> This is the core of our motivation, supported by a thorough empirical study in Section 2. The reviewer is correct that using a larger batch size would indeed improve hardware utilization; however, it would result in a suboptimal model. We conduct a comprehensive empirical study to identify a smaller batch size as a core motivation for our work (Section 2). As we demonstrate in our Observation #3 and as supported by prior works [1][2][3], LoRA fine-tuning requires small batch sizes (e.g., 1, 2, 4) to achieve optimal model quality.  Regarding LoRA ranks, as shown in Table 1, our search space already includes higher ranks (up to 128). Even with large LoRA ranks, hyperparameter tuning remains necessary.
>
> [1] LoRA Land: 310 Fine-tuned LLMs that Rival GPT-4, Zhao et. Al. https://arxiv.org/abs/2405.00732
>
> [2] A Note on LoRA. Fomenko et. al. https://arxiv.org/abs/2404.05086
>
> [3] LoRA Without Regret. Schulman et. al. https://thinkingmachines.ai/blog/lora/
>
> **(W2, Q1) Total efficiency gain is unclear**
>
> With lots of pre-trained models available on platforms such as Hugging Face, we target the use case where a pre-trained model is available and users wish to adapt it to their own use cases (LoRA fine-tuning as a service) [1]. What we report is the direct and practical efficiency gain. For a user who wishes to deploy LoRA adapters on a model, they must undergo HPO to find the ideal LoRA. In our case, HPO is not a fraction of the total training procedure but the workload itself. PLoRA directly enables efficient concurrent tuning of LoRA adapters. Our reported gains are the direct gains in fine-tuning efficiency.
>
> [1] LoRA Without Regret. Schulman et. al. https://thinkingmachines.ai/blog/lora/
>
>
> **(Q2) Can PLoRA be extended to LoRA variants?**
>
> Yes, PLoRA can be directly applied to LoRA variants such as QLoRA (base-model weight quantization), DoRA (which decomposes base-model weights), as they only change the base-model weights and do not affect the LoRA computations. PLoRA is also compatible with variants such as AdaLoRA, which uses heterogeneous ranks for different layers. In our framework, this could be easily supported by specifying different ranks for different layers.

---

> ### Comment · Reviewer_nryQ · 2025-11-23
> **Thanks for the response.**
>
> Thanks for the detailed response. I have no further questions and would like to maintain the positive rating.

---

### Official Review · Reviewer_8fpn · 2025-10-25

**Soundness:** 2
**Presentation:** 2
**Contribution:** 2
**Rating:** 4
**Confidence:** 4

**Summary:**

This paper introduces PLoRA, a system designed to accelerate the hyperparameter tuning process for LoRA by addressing hardware underutilization. The authors first provide an empirical study demonstrating the necessity of LoRA hyperparameter tuning and identifying that typical tuning jobs, which often use small batch sizes, lead to inefficient GPU usage. To solve this, PLoRA proposes packing multiple LoRA configurations into a single fine-tuning job. The system comprises an offline packing planner, which uses an optimization algorithm to schedule jobs, and an online execution engine equipped with custom GPU kernels for efficient computation of packed adapters. Experimental results show significant reductions in the overall tuning time (makespan) and improvements in training throughput.

**Strengths:**

1. The paper begins with a thorough empirical study (Section 2) that clearly establishes the problem's significance. By demonstrating that optimal LoRA hyperparameters are task- and model-specific and that fine-tuning often leads to hardware underutilization, the authors provide a compelling justification for their work.

2. PLoRA is a well-designed, end-to-end system that combines high-level scheduling with low-level kernel optimizations. The two-stage approach of an offline planner and an online execution engine is practical, and the development of custom packed LoRA kernels directly addresses the core performance bottleneck. The reported improvements in makespan and throughput are substantial and demonstrate the effectiveness of the proposed approach.

**Weaknesses:**

1. The proposed packing strategy assumes that all configurations within a job are trained for the same duration. This is incompatible with adaptive HPO algorithms like HyperBand or Asynchronous Successive Halving (ASHA), which rely on early termination of unpromising trials to improve efficiency.

2. The scheduler appears to assume that all LoRA configurations in the search space require the same number of training steps. In practice, different configurations (e.g., with different learning rates) may converge at different speeds, or a user might want to train them for different numbers of epochs.

3. The paper compares against "Min GPU" and "Max GPU" baselines, which represent simple, manual strategies. While reasonable, they do not represent more sophisticated scheduling heuristics. The contribution of the ILP-based planner could be better isolated by comparing it against a simpler, greedy packing algorithm.

4. The planner relies on a recursive algorithm (DTM) that calls an ILP solver. While the paper states the offline planning time is negligible for 120 configurations, this approach may not scale to scenarios with thousands of configurations, which are common in large-scale HPO.

5. The paper claims applicability to other parallelism strategies like FSDP and provides a formulation in the appendix. However, all experiments are conducted with Tensor Parallelism (TP). FSDP, particularly ZeRO-3, has fundamentally different memory and communication patterns that might complicate the packing of heterogeneous LoRA adapters.

6. The planner's decisions are based on a cost model that estimates throughput from the first few training iterations.

7. The custom CUDA kernels are tuned for specific GPU architectures (Ampere). This is a common practice for high-performance systems but limits portability and may require significant engineering effort for new hardware.

8. The main text presents a simplified throughput maximization problem (Eq. 1), while the appendix details a full makespan minimization MILP (Eq. 8). The connection and transition between these two formulations could be made clearer.

**Questions:**

1. How can the PLoRA framework accommodate adaptive HPO strategies that require early stopping? Would it be possible to dynamically dissolve a packed job or stop gradient updates for certain adapters within a pack if they are identified as unpromising?

2. How does the planner handle a search space where configurations have heterogeneous training-length requirements? Does the current model assume a fixed number of steps for all trials, and what would be the implications of relaxing this assumption?

3. The paper compares against "Min GPU" and "Max GPU" baselines, which represent simple, manual strategies. While reasonable, they do not represent more sophisticated scheduling heuristics. The contribution of the ILP-based planner could be better isolated by comparing it against a simpler, greedy packing algorithm.

**Details Of Ethics Concerns:**

Null.

---

> ### Author Response · Authors · 2025-11-21
>
> We sincerely thank the reviewer for the review, for pointing out the thoroughness of our empirical study, and for recognizing that we provide a well-designed end-to-end system. We will address your concerns below and hope this can earn your full support for the paper.
>
> **(W1, W2, Q1, Q2) Compatibility with adaptive HPO and heterogeneous training lengths.**
>
> We first want to clarify that our work is orthogonal but complementary to HPO search strategies. PLoRA is an execution engine that solves the problem: "Given a set of N configurations, how do we execute them in the minimum amount of time?" Before our work, the answer was "run them one by one," which (as we show in Section 3) is extremely inefficient. PLoRA provides the first high-performance engine for this task.
>
> Our current fixed-duration job model is a simplification, albeit a very reasonable one. For instance, many HPO methods, such as SHA, Hyperband, and BO, follow this model. Stopping gradient updates would be easy to implement, but variable-duration jobs would add a layer of complexity to the already complex scheduling problem. We believe this could be of independent research interest. We aim to be the first work to unleash large-scale and concurrent LoRA fine-tuning.
>
> **(W3, Q3) Baseline simplicity.**
>
> MinGPU is exactly how a practitioner would approach this problem today. Users deploy it on the minimum number of GPUs that does not overflow the memory.
>
> The advantage of our planner lies in its holistic, joint optimization of both packing and parallel strategies. If we have an Oracle packer and a MinGPU strategy, greedy scheduling may work well when the model is a tiny fraction of GPU memory (e.g., 3B models on A100), but may lead to inefficiencies otherwise (e.g., 14B models on A100). Note that this decision would also change when we use the same base model for different LoRA adapters (e.g., LoRA rank 8 and 128 would have vastly different memory costs). The DTM planner jointly optimizes both distributed training configurations and adaptive packing to trade off the different communication and distributed training overhead from different parallelization strategies.
>
> **(W4) DTM planner scalability to thousands of configurations.**
>
> We respectfully disagree and show below that this is not a practical bottleneck. Our reasoning is twofold:
>
> **1. PLoRA Enables Large-Scale HPO.**
>
> The reviewer argues that some HPO sweeps involve thousands of configurations. However, without PLoRA, a user would need to launch thousands of sequential, hardware-underutilizing jobs, which is practically infeasible. PLoRA's core contribution is transforming this problem. By supporting massive concurrent training, our work is the first to make such a large-scale sweep feasible. A user can now explore 1000s of configurations because of PLoRA.
>
> **2. Planner Overhead as a percentage stays constant even at scale.**
>
> For a 1000+ configuration sweep (which was enabled via PLoRA), our planner's overhead remains negligible. The total planning time is a function of the number of jobs (and thus the number of iterations in the DTM algorithms), which scales linearly with the number of LoRA configurations. The only remaining concern is whether the time required to plan a single job per DTM algorithm iteration increases as the pool of available configurations K grows. Modern ILP solvers like Gurobi are designed to handle optimization problems with 10^7 or more variables. An ILP instance with O(10k) variables (in our case, whether to include a configuration into the next job is a binary variable) is well within the capabilities of modern ILP solvers.
>
> To provide concrete evidence, we measured the time our ILP solver (Gurobi) takes to solve a single packing instance (Line 6) as we increase the number of LoRA configurations.
>
> | Number of LoRA adapters |  120  |  1200  |  12000  |
> |-----|-----|-----|-----|
> | Gurobi solver time per iteration |  0.214s  |  0.215s  |  0.215s  |
>
> As we see from the table above, the solver can handle 12K LoRA configurations in the same time as 120 configurations; therefore, the percentage overhead of our planning algorithm would remain constant as the number of LoRA adapters increases, in addition to the fact that PLoRA enables such massive-scale concurrent training in the first place.
>
> **(W5) Applicability to FSDP.**
>
> We believe this is a strength of our paper. The generality of PLoRA’s job planner is defined by its ability to model different hardware/parallelism constraints. Therefore, we include how the optimization formulation would change under different hardware/parallelism constraints. In Appendix B, we explicitly provide the exact MILP constraint equations for ZeRO-1/2/3. This demonstrates that our framework can target FSDP. All that is required is to use the constraints we provide in the appendix instead. This shows the generality and flexibility of our system design.

---

> ### Author Response · Authors · 2025-11-21
>
> **(W6) Planner decisions based on a cost model from the first few iterations.**
>
> We argue that this is a strength and standard practice for systems-level performance modeling. Profiling a few warm-up iterations is the standard, rigorous, and low-cost (reusable for the same base model) method for building performance models. Throughput is a stable property of a training job after initialization. This methodology is precisely what enables our planner to be fast and accurate. For instance, as a standard practice, we profile 10 iterations to obtain stable readings. This is <1% of the total training cost, as the fine-tuning itself often exceeds 1000 iterations.
>
> **(W7) Custom CUDA kernels limit portability.**
>
> We believe this is also a key strength of our paper. Any high-performance library (e.g., PyTorch, vLLM, CUTLASS) relies on hardware-tuned kernels, and PLoRA provides this full-stack contribution: from the high-level ILP (Eq. 8) and scheduler (Alg. 2) to the low-level execution engine and kernels (Appendix C).
>
> We demonstrate the principle is portable by showing high performance on both A100 (Table 7) and A10 (Table 8) GPUs. While one may need to tune kernel parameters, which can be hardware-specific (as they are in all ML libraries), the kernel design and packing strategies are general.
>
> **(W8) The connection between Eq. 1 and Eq. 8 is unclear.**
>
> We thank the reviewer for the suggestion, and we will update the writing. We explain the relationships in Lines 273-293. We will add the following paragraph to make the transition clearer: Our goal is to solve the makespan problem (Eq. 8). We approximate this by maximizing instantaneous throughput (Eq. 1), which we solve with DTM (Alg. 1) and prove is near-optimal (Appendix D).

---

### Meta-Review · Area_Chair_E2Z3 · 2026-01-12

**Summary:**

It received ratings of 4,6,6. Two reviewers replied to the rebuttal and said they won't change the rating and I do not think the other reviewer would have changed either. Overall, I believe some concerns that are pointed out bellow are not fully addressed.

**Reviewer Concerns:**

The main concerns include making the approach incompatible with adaptive HPO methods and heterogeneous convergence behaviors; limited and overly simplistic baselines that do not adequately isolate the benefit of the ILP-based planner or compare against stronger scheduling and utilization strategies; scalability concerns arising from the recursive ILP-based planning, which is only evaluated on small search spaces and may not scale to large HPO workloads; limited validation of generality, as all experiments are restricted to tensor parallelism despite claims of applicability to FSDP; and unclear practical impact, since the reported efficiency gains focus on the hyperparameter tuning phase, which may represent only a small fraction of total training time, leaving end-to-end benefits insufficiently demonstrated.

**Reviewer Scores:**

It received ratings of 4,6,6. The authors provided rebuttal. The two 6 ratings responded and said they won't change the score. The first review (rating of 4) did not respond. Reading the author response, while it addresses some of the concerns, there are still some concerns that are not fully addressed. First, PLoRA’s execution model still assumes fixed-duration, synchronized jobs, making it incompatible with adaptive HPO methods such as ASHA or HyperBand that rely on early stopping and variable budgets. The response acknowledges this limitation but does not explain how packed jobs could be dynamically adjusted in practice. Second, the evaluation lacks a simple greedy or heuristic packing baseline. While the MinGPU and MaxGPU baselines reflect common usage, they do not isolate the benefit of the ILP-based planner, leaving it unclear whether similar gains could be achieved with simpler scheduling strategies.

Hence, I do not think that reviewer (rating of 4) would have changed their score either after reading the rebuttal.

---

### Decision · Program_Chairs · 2026-01-26

Reject